

# Occurrence of concurrent infections with multiple serotypes of dengue viruses during 2013–2015 in northern Kerala, India

Manchala Nageswar Reddy, Ranjeet Dungdung, Lathika Valliyott and Rajendra Pilankatta

Department of Biochemistry and Molecular Biology, Central University of Kerala, Kasargod, Kerala, India

Corresponding author
Rajendra Pilankatta,
praj74@cukerala.ac.in

## ABSTRACT

**Background**. Dengue is a global human public health threat, causing severe morbidity and mortality. The occurrence of sequential infection by more than one serotype of dengue virus (DENV) is a major contributing factor for the induction of Dengue Hemorrhagic Fever (DHF) and Dengue Shock Syndrome (DSS), two major medical conditions caused by DENV infection. However, there is no specific drug or vaccine available against dengue infection. There are reports indicating the increased incidence of concurrent infection of dengue in several tropical and subtropical regions. Recently, increasing number of DHF and DSS cases were reported in India indicating potential enhancement of concurrent DENV infections. Therefore, accurate determination of the occurrence of DENV serotype co-infections needs to be conducted in various DENV prone parts of India. In this context, the present study was conducted to analyse the magnitude of concurrent infection in northern Kerala, a southwest state of India, during three consecutive years from 2013 to 2015.

**Methods**. A total of 120 serum samples were collected from the suspected dengue patients. The serum samples were diagnosed for the presence of dengue NS1 antigen followed by the isolation of dengue genome from NS1 positive samples. The isolated dengue genome was further subjected to RTPCR based molecular serotyping. The phylogenetic tree was constructed based on the sequence of PCR amplified products.

**Results**. Out of the total number of samples collected, 100 samples were positive for dengue specific antigen (NS1) and 26 of them contained the dengue genome. The RTPCR based molecular serotyping of the dengue genome revealed the presence of all four serotypes with different combinations. However, serotypes 1 and 3 were predominant combinations of concurrent infection. Interestingly, there were two samples with all four serotypes concurrently infected in 2013.

**Discussion**. All samples containing dengue genome showed the presence of more than one serotype, indicating 100% concurrent infection. However, the combination of serotypes 1 and 3 was predominant. To the best of our knowledge, this is the first report indicating the concurrent infection of dengue in the northern Kerala, India. The phylogenetic analysis of dengue serotype 1 identified in this study shows a close relationship with the strain isolated in Delhi and South Korea during the 2006 and 2015 epidemics respectively. Similarly this study indicates that the phylogeny of dengue serotype 3 of northern Kerala is more closely related to dengue isolate of Rajasthan state, India. The geographical and climatic conditions of Kerala favours the breeding of

both the mosquito vectors of dengue (*Aedes albopictus* and *Aedes aegypti*), which may enhance the severity of dengue in the future. Therefore, the study provides an alarming message for the urgent need of an antiviral strategy or other health management systems to curb the spread of dengue infection.

## INTRODUCTION

Dengue virus (DENV) belongs to the family of Flaviviridae and genus *Flavivirus* and poses a global threat resulting in significant morbidity and mortality. The virus is transmitted by day-biting mosquito, *Aedes aegypti* (*Liu-Helmersson et al., 2014*). However, there is no vaccine or antiviral drug available that can neutralize all the four serotypes of dengue viruses.

There are four distinct DENV1-4 serotypes circulating all over the world and causing DENV infection. The infection causes symptoms ranging from acute febrile illness to severe manifestations, including bleeding and organ failure resulting in the DHF or DSS (*Gubler, 1998*; *Moi, Takasaki & Kurane, 2016*). Co-infection with circulating DENV 1 and DENV 2 was reported in 1982 in Columbia (*Gubler et al., 1985*). It has been known that sequential infection of more than one serotype of dengue increases the severity of dengue symptoms (*Hammon, 1973*). Meanwhile, there are reports indicating concurrent infection of dengue with more than one serotype (*Anoop et al., 2010*). However the correlation between concurrent infection of dengue with more than one serotype and severity of the disease symptoms is not well established. In this context, the current study becomes highly relevant and gives a platform for future investigation to understand the severity of the disease and concurrent infection caused by different dengue serotypes.

In the last 50 years, co- circulation of dengue serotypes was reported in South Asia, including India. The first virologically confirmed dengue case was reported in the east coast of Calcutta, India during 1963–64 (*Carey et al., 1966*; *Sarkar et al., 1964*). In addition, a dengue outbreak at Kanpur, India was documented during 1968 by DENV 4 (*Chaturvedi et al., 1970*). The presence of DENV 3 was found in patients as well as *A. egypti* mosquitoes in Vellore, India in 1966, and since then all the four types of DENV have been co circulated and isolated from patients and mosquitoes (*Myers & Carey, 1967*; *Wenming et al., 2005*).

In 1996, DENV 2 serotype infections were noticed in India, followed by spreading all over the country (*Shah, Deshpande & Tardeja, 2004*; *Singh et al., 2000*). The capital city of India, Delhi, became hyperendemic by hosting all four dengue virus serotypes by 2003 (*Dar et al., 2003*) with the coinfection of DENV 1 and DENV 3 in 2005 (*Gupta et al., 2006*). The magnitude of concurrent infection (19%) observed during the Delhi outbreak in 2006 is much higher in comparison with Taiwan (9.5%) and Indonesia (11%). Furthermore, replacement of DENV 2 and 3 with DENV 1 as the predominant serotype in Delhi over a period of three years (2007–2009) has been reported.

The occurrence of dengue fever was reported in the Kottayam district of Kerala, a south-western region of India, followed by an outbreak in 2003. Concurrent infection with all three DENV 1–3 were reported in a large number of patients in 2008 (*Anoop et al., 2010*) in Ernakulum district, a central region of Kerala. In 2013, various parts of India, including Davangere and Central Karnataka out of 123 positive NS1 antigen samples 56(45.5%) were infected with dengue fever, 37(30.1%) with DHF, and 30(24.4%) with DSS (*Kalappanvar et al., 2013*).

Since dengue cases are increasing at an alarming rate and causing a major health threat in tropical countries, it is necessary to identify and confirm the viral serotypes through epidemiological surveillance studies. The present study was conducted in order to specifically identify dengue virus and to assess the concurrent infection in the northern Kerala (Malabar region), India during three consecutive years (2013–2015). Some of these areas are ideal ecosystems for the proliferation of *Aedes albopictus* mosquitoes. In this study, NS1 positive samples were further tested by amplification of the junction region of capsid and pre membrane gene (CprM) of the dengue viral genome (*Lanciotti et al., 1992*) by single step reverse transcriptase polymerase chain reaction (RT-PCR) followed by specific serotype identification.

To the best of our knowledge, we are reporting for the first time the 100% concurrent infection by different dengue serotypes among the samples analysed, including two samples which were positive for all four dengue serotypes.

# PATIENTS AND METHODS

## Sample collection

The viremic blood samples from patients with suspected dengue symptoms (as per WHO guidelines) were collected from the Government hospital and local diagnostic centres located in the northern Kerala, India. The Institutional Ethics committee (IEC) clearance (O.R.No:IAD/IEC/13/14) was taken from the Institute of Applied Dermatology (IAD), Kasaragod district, Kerala for conducting this study. The prior informed consent was obtained from all participating human subjects. The blood samples used in this study were collected between June and August of 2013, 2014, and 2015, respectively. The blood samples were collected from patients who came to the diagnostic centres with a doctor's prescription or with the suspicion (1–5 days). Donors were aged 5–60 years of either sex, including paediatrics (0–5 years old). Two millilitres of blood was collected in a vacutainer tube from each individual and serum was separated by centrifugation as per the standard procedure.

## Detection of NS1 antigen by capture ELISA

The presence of NS1antigen in the patient serum sample was screened for using a DENV specific NS1 detection Enzyme Linked Immunosorbent Assay (ELISA) kit (Jmitra and Co., New Delhi, India). The kit contains monoclonal antibodies against NS1 coated on micro wells, which can detect NS1 antigen secreted by DENV in the infected patient. Fifty microliters of serum sample per micro well was used in the 96 well plate assay. Normal

serum specimens obtained from healthy humans of same sex and age groups were included as controls.

## Isolation of dengue viral RNA

Dengue viral RNA was extracted from NS1 positive serum samples using the pure link RNA mini kit (Invitrogen, Carlsbad, CA, USA) according to the manufacturer's instructions. Clarified human serum sample of 150 microliters volume was used for RNA isolation. All RNA samples were examined for their purity and concentration using Nano photometer (2000C; Thermo Scientific, Waltham, MA, USA).

## Amplification of dengue viral CprM region by RT-PCR

Isolated Dengue viral RNA from serum samples was subjected to single step reverse transcription polymerase chain reaction (RT-PCR) (*Lanciotti et al., 1992*). Ten microliters of PCR amplified product was analysed on a 1% agarose gel and the size compared with a 1 Kb plus DNA ladder.

## Nested PCR

A second round of amplification was initiated using one microliter of the above PCR product (1:100 in sterile distilled water) as a template in the subsequent nested PCR reaction. The reaction mixture contained all the components necessary for PCR amplification including D1 as a forward primer and dengue virus type-specific reverse primers TS1, TS2, TS3, and TS4 in separate individual tubes. The CprM consensus regions and each serotype-specific primers were designed as mentioned by *Lanciotti et al. (1992)* with slight modification of TS4 primer 5′-CTCTGTTGTCTTAAACAAGAGA-3′. The PCR amplified products were analysed by electrophoresis on 1% agarose gel.

## Nucleotide sequencing

All necessary Good Lab Practices (GLP) were employed to avoid artifacts. The precautions were taken to avoid the contamination of serum samples by bar coding of samples. The cross contamination was prevented by setting of the PCR independently in the separate tubes with pair of specific primers. The tube containing no template was also setup as a negative control. The RT-PCR product (511 bp) obtained using D1 and D2 primers was separated on the 1% agarose gel. The DNA band was eluted from the gel using Wizard gel and PCR clean up system (Promega, Madison, WI, USA) as per the manufacturer instructions and was sequenced directly using big dye terminator V3.1 ready reaction sequencing mixture in an automated AB3500 Genetic Analyzer (Applied Biosystems, Foster City, CA, USA). Additionally, nested PCR products obtained using D1 and TS1/TS3 dengue virus serotype specific primers were purified and sequenced as detailed above.

## Amino acid sequence similarity and diversity

The nucleotide sequences obtained from the above method were submitted to GenBank database (NCBI). These sequences were translated using the Expasy translation (EMBL) tool. These homologues amino acid regions were aligned with the partial or full length amino acid sequences of dengue isolates of diverse geographical locations (KP406801,

EF127001, DQ285562, JQ922545, JN903581, KM403635, KR024707, KT187563, KP723473, JQ917404, JN713897) retrieved from GenBank, using BioEdit sequence alignment editor.

## Molecular phylogenetic analysis of DENV 1 and DENV 3 by Maximum Likelihood method

The phylogenetic analysis was conducted independently for DENV 1 (GenBank accession no. KJ954284) and DENV 3 (GenBank accession no. KM042094) of northern Kerala with the gene sequences of dengue virus isolates of different locations available in NCBI GenBank. The percentage of the bootstrap supporting values was shown at major nodes on the tree. The tree is drawn to scale, with branch lengths measuring the number of substitutions per site. The reliability of the analysis was evaluated by a bootstrap test (MEGA), Tamura-Nei model (*Tamura et al., 2012*). All positions containing gaps and missing data were eliminated.

## RESULTS

There were, in total, 120 serum samples of dengue infected individuals analysed between 2013 and 2015. Among these 120 samples, a total of 62 (51%) were male, 35 (29%) female, and 23 (20%) paediatric (0–5 years old) dengue cases (Fig. 1). Both males and females belong to 30–40 years age group with an approximate ratio of 2:1. This ratio is similar with the previous reported value (*Mishra et al., 2015*). There were 23 (20%) pediatric (0–5 years old) cases, consisting of 14 males and nine females. Out of 120 serum samples, 100 (83%) were found to be NS1 positive, indicating the possibility of finding dengue viral RNA genome.

## Dengue viral genome isolation and serotyping

Dengue viral RNA was isolated from NS1 positive serum samples, as detailed in the patients and methods section. The isolated RNA was subjected to RT-PCR based amplification (Fig. 2) using D1 and D2 primers. There was dengue specific amplified product of size 511 bp present only in 26 samples. This observation confirms the presence of dengue genome in these 26 samples out of 100 samples, which were NS1 positive. The initial dengue NS1 antigen detection and the presence of identifying dengue genome in the serum samples indicates the possibility of dengue infection. A representative gel image, which indicates the presence of 511 bp amplified product is given in Fig. 3A. Out of fifty-one NS1 positive samples in 2013, thirteen samples were found to be positive for dengue genome. Similarly, there were three out of seventeen and ten out of thirty two samples positive for dengue genome analysed during 2014 and 2015, respectively (Table 1). All samples were co-infected with more than one serotype with various combinations of serotypes as given in Table 1 and Figs. 3B–3D.

## Concurrent infection

The above amplified RTPCR product (511 bp) was used as a template for nested PCR, using serotype specific primers as detailed in methods section. The nested PCR analysis of dengue serotyping showed that a single individual is hosting for more than one dengue virus serotype circulating in the blood. In all 26 samples of multiple infection, three different
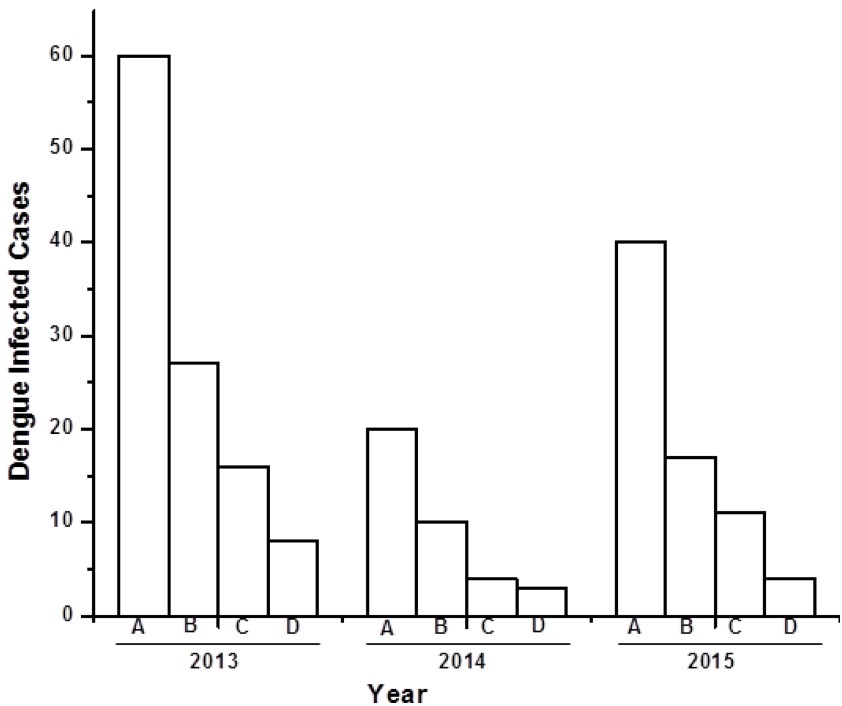

**Figure 1** **Bar diagram showing the number of dengue virus infected cases under different sex group in three consecutive years (2013–2015).** $X$-axis represents year of study and $Y$-axis represents number of dengue infected cases. (A) Total number of dengue infected individuals. (B–D) Dengue NS1 positive male, female and pediatrics respectively.

**Table 1** **Dengue infected samples analysed during 2013–2015.** Total 120 samples were analyzed during 2013, 2014 and 2015, out of which 100 samples were NS1 positive. Among the 100 samples, 26 samples were dengue viral RNA positive, out of which, three serotypes (DENV 1, 2 and 3) co-existed in eight samples, two serotypes (DENV 1 and DENV 3) co-existed in nine samples and two serotypes (DENV2 and DENV3) co-existed in seven samples and four co serotypes (DENV1, 2, 3 and 4) co-existed in two samples. In all cases, multiple infections were observed. However, concurrent infection of all four serotypes was only observed during 2013.

| S.No | Sample description | 2013 | 2014 | 2015 |
|------|--------------------|------|------|------|
| 1 | Total number of samples | 60 | 20 | 40 |
| 2 | NS1 positive | 51 | 17 | 32 |
| 3 | NS1 negative | 9 | 3 | 8 |
| 4 | RNA positive | 13 | 3 | 10 |
| 5 | Single infection | Nil | Nil | Nil |
| 6 | Multiple infection | All | All | All |
| 7 | DENV1 and DENV3 | 4 | 1 | 4 |
| 8 | DENV2 and DENV3 | 2 | 1 | 4 |
| 9 | DENV1,2 and 3 | 5 | 1 | 2 |
| 10 | DENV1,2,3 and 4 | 2 | 0 | 0 |

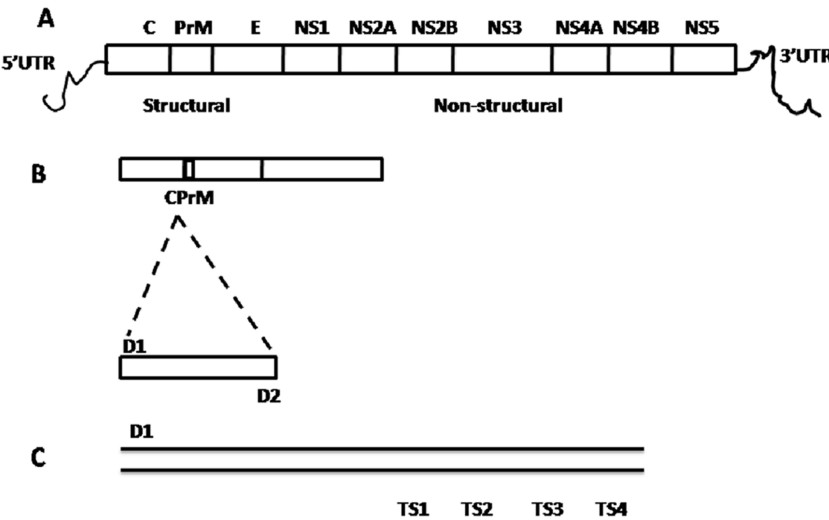

**Figure 2** Schematic representation of dengue viral CprM junction amplification by RT-PCR. (A) Dengue viral RNA has 5′ UTR region, three structural (capsid (C), pre membrane (prM), and envelope (E)) and seven nonstructural (NS1, NS2A, NS2B, NS3, NS4A, NS4B, and NS5) genes ends with 3′ UTR region. (B) CPrM junction (511 bp) of structural region was amplified with D1 and D2 primers using dengue viral RNA as a template. (C) The amplified products with D1 and sero specific TS1, TS2, TS3 and TS4 primers independently, indicates the expected size (bp) 482, 119, 290 and 392 respectively by nested PCR.

serotypes co-existed in eight samples, two different serotypes co- existed in sixteen samples, and four different serotypes co-existed in two samples. There are nine cases of concurrent infection with DENV1 and DENV 3, seven cases with DENV 2 and DENV 3, eight cases with DENV 1, 2 and 3, and two cases with DENV 1, 2, 3 and 4 combinations as shown in the Table 1. There was a large number of concurrent infection with DENV 1 and DENV 3. The concurrent infection of all four serotypes is an alarming indication and needs to be investigated in detail. The sequential infection with more than one serotype of a particular region leads to a severe cause for eliciting antibody dependent immune response.

## Nucleotide sequencing and dengue virus serotyping

The D1 and D2 primer based amplified PCR product corresponding to CPrM region was sequenced (GenBank accession no. KX031992) and the data confirmed the presence of dengue viral genome sequence in the samples. The nucleotide sequence was found to be 99% similar with the existing CprM region of DENV 3 viral strains in GenBank database. Blast analysis, using CprM junction nucleotide sequence, showed similarity with DENV 3 isolates from Pakistan with 99% query coverage (GenBank accession no. KF041254). Nucleotide sequencing of DENV 1 specific nested PCR product, obtained using D1 and TS1 primer combinations, gives the size of 423 bp (GenBank accession no. KJ954284). The corresponding amino acid sequence (1-139 aa) derived from DEV 1 specific amplified region represents non functional poly protein partial sequence. DENV 3 specific nested PCR product using D1 and TS3 primer combinations gives the size of 235 bp (GenBank accession no. KM042094) on a 1% agarose gel. However, PCR amplified product of DENV 2 and DENV 4 were the lesser in quantity, the nucleotide sequences were not obtained with clarity.

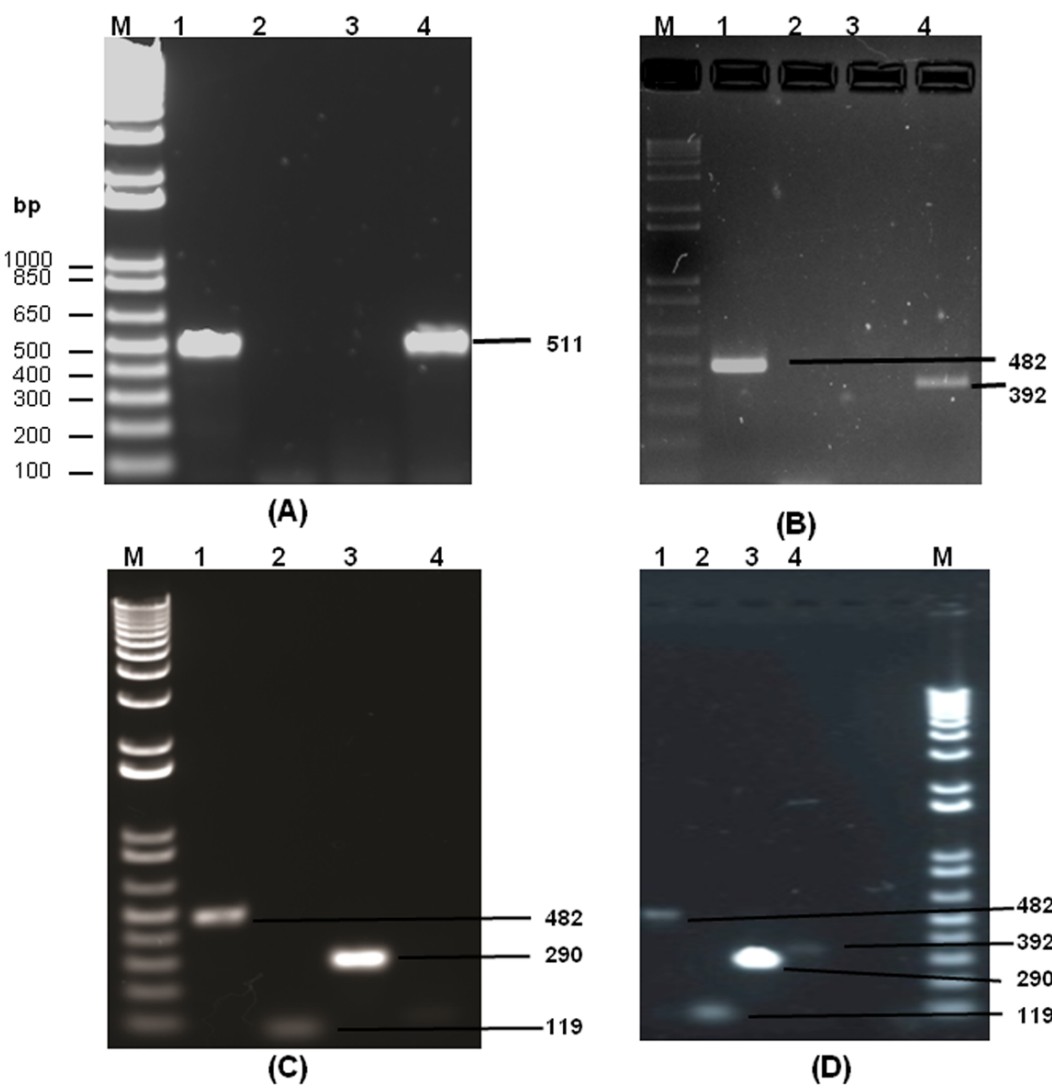

**Figure 3** **The agarose gel images (1%) showing the amplified DNA from dengue viral genome using RT-PCR.** (A) shows a representative DNA gel showing the 511 bp DNA fragment obtained by RT-PCR using dengue viral RNA as template with D1 (forward) and D2 (reverse) primers. Lane 1 and Lane 4 shows the presence of dengue viral genome in the sample as evidenced by a clear DNA band of 511 bp size. However, Lane 2 and 3 did not show the presence of dengue viral genome in the corresponding samples. (B–D) represents DNA gel images showing the nested PCR products by serotype specific primers with D1&TS1 (Lane 1), D1&TS2 (Lane 2), D1&TS3 (Lane 3), D1&TS4 (Lane 4) primer combinations using 511 bp PCR product as a template. In (B), Lane 1 and 4 shows the presence of amplified product of size 482 bp and 392 bp respectively indicating the presence of DEV 1 and 4 in the same individual. Similarly, Figure C Shows the presence amplified product of size 482 bp, 119 bp and 290 bp in Lane 1, 2 and 3 respectively, indicating the concurrent infection of dengue serotype 1, 2 and 3 in the same individual. Interestingly, (D) indicates the concurrent infection of all four serotypes in the same individual as evidenced by the amplified product of size 482 bp, 119 bp 290 bp and 392 bp in all four lanes respectively. M, indicates 1 kb DNA ladder (bp of ladder size is shown in (A), and lane number is indicated at the top of each figure part).

## Amino acid sequence diversity

To observe any mutations, the amino acid sequence of DENV1 was compared with the other closely related DENV 1 strains. The comparison of the amino acid sequence (1-139 aa) of DENV 1 (CUKKEL201308001; GenBank accession no. KJ954284) with various DENV 1 isolates existing in the data base (GenBank accession no KP406801, EF127001, DQ285562, JQ922545, JN903581, KM403635, KR024707, KT187563, KP723473, JQ917404, JN713897) revealed an identity of 98%–99%, except for Valine (V) which is replaced by Isoleucine (I) at 131st position as shown in Fig. 4. However, replacement of Isoleucine with Valine was reported at residue 106 of the capsid protein in the isolates obtained from the 1997, 1998 and 2001 outbreaks in the Caribbean islands pertaining to the Asian/American genotype (*Gardella-Garcia et al., 2008*).

## Phylogenetic analysis of DENV 1 and DENV 3

In the molecular phylogenetic analysis (Fig. 5) by Maximum Likelihood method, the bootstrap support values indicates that the sequence of DENV 1 isolate (CUKKEL201308001; GenBank accession no. KJ954284) shares a common ancestor relationship with DENV 1 isolate (GenBank accession no. KP406801) and DENV 1 isolate 06/1/del2006 (GenBank accession no. EF127001) among 28 nucleotide sequences used for the analysis (Fig. 5). Similarly the phylogenetic analysis (Fig. 6) of dengue 3 virus isolate of Northern Kerala (CUKKEL2013/02; GenBank accession no. KM042094) indicates that this strain is closely related to Dengue virus isolate (DENV-3/DMRC/Bal87/2013) polyprotein gene, partial cds (KT239735) of Rajasthan, India.

## DISCUSSION

The surveillance of concurrent infection of multiple dengue serotypes as well as its co circulation becomes extremely important for understanding viral pathogenesis and also for developing a dengue tetravalent vaccine able to neutralize all four serotypes.

There are only a few independent studies conducted on co-circulation and concurrent infection of dengue in India. These studies include the report on the concurrent infection of dengue which was observed in Delhi in 2006 (*Bharaj et al., 2008*). In this outbreak, nine out of 48 samples (19%) were identified positive for dengue virus as a concurrent infection with more than one dengue virus serotype. The concurrent infection with the involvement of three dengue viral serotypes (DENV 1, 2, and 3) at an approximate rate of 56.8% was observed in Ernakulam, Kerala state in 2008 (*Anoop et al., 2010*). A large outbreak of dengue fever (DF) was reported in 2007, in the locality of Indo-Myanmar boarder with the co circulation of concurrent infection of DENV 2 & 3, 1 & 3, and 1 & 4 serotypes (*Khan et al., 2013*).

In 1980, DENV 1 and 2 were first observed in Columbia. By 1994, many parts of the world, including, Bolivia, Brazil, Costa Rica, El Salvador, French Guiana, Guatemala, Honduras, Mexico, Nicaragua, Panama, Peru, Puerto Rico, Trinidad and Tobago, and Venezuela were hosting all four dengue serotypes (*Lorono-Pino et al., 1999*). The first case of a dual infection with two dengue virus serotypes (DENV 1 and DENV 4) was reported in the serum of a 16-year-old male during the 1982 outbreak in Puerto Rico (*Gubler et al., 1985*).

```
            ....|....|  ....|....|  ....|....|  ....|....|  ....|....|
                10          20          30          40          50
KJ954284    MLKRARNRVS  TGSQLAKRFS  KGLLSGQGPM  KLVMAFIAFL  RFLAIPPTAG
KP406801    MLKRARNRVS  TGSQLAKRFS  KGLLSGQGPM  KLVMAFIAFL  RFLAIPPTAG
EF127001    MLKRARNRVS  TGSQLAKRFS  KGLLSGQGPM  KLVMAFIAFL  RFLAIPPTAG
DQ285562    MLKRARNRVS  TGSQLAKRFS  KGLLSGQGPM  KLVMAFIAFL  RFLAIPPTAG
JQ922545    MLKRARNRVS  TGSQLAKRFS  KGLLSGQGPM  KLVMAFIAFL  RFLAIPPTAG
JN903581    MLKRARNRVS  TGSQLAKRFS  KGLLSGQGPM  KMVMAFIAFL  RFLAIPPTAG
KM403635    MLKRARNRVS  TGSQLAKRFS  KGLLSGQGPM  KMVMAFIAFL  RFLAIPPTAG
KR024707    MLKRARNRVS  TGSQLAKRFS  KGLLSGQGPM  KMVMAFIAFL  RFLAIPPTAG
KT187563    MLKRARNRVS  TGSQLAKRFS  KGLLSGQGPM  KMVMAFIAFL  RFLAIPPTAG
KP723473    MLKRARNRVS  TGSQLAKRFS  KGLLSGQGPM  KMVMAFIAFL  RFLAIPPTAG
JQ917404    MLKRARNRVS  TGSQLAKRFS  KGLLSGQGPM  KMVMAFIAFL  RFLAIPPTAG
JN713897    MLKRARNRVS  TGSQLAKRFS  KGLLSGQGPM  KLVMAFIAFL  RFLAIPPTAG

            ....|....|  ....|....|  ....|....|  ....|....|  ....|....|
                60          70          80          90          100
KJ954284    ILARWSSFKK  NGAIKVLRGF  KKEISSMLNI  MNRRKRSVTM  LLMLLPTALA
KP406801    ILARWSSFKK  NGAIKVLRGF  KKEISSMLNI  MNRRKRSVTM  LLMLLPTALA
EF127001    ILARWSSFKK  NGAIKVLRGF  KKEISSMLNI  MNRRKRSVTM  LLMLLPTALA
DQ285562    ILARWSSFKK  NGAIKVLRGF  KKEISSMLNI  MNRRKRSVTM  LLMLLPTALA
JQ922545    ILARWSSFKK  NGAIKVLRGF  KKEISSMLNI  MNRRKRSVTM  LLMLLPTALA
JN903581    ILARWSSFKK  NGAIKVLRGF  KKEISSMLNI  MNRRKRSVTM  LLMLLPTALA
KM403635    ILARWSSFKK  NGAIKVLRGF  KKEISSMLNI  MNRRKRSVTM  LLMLLPTALA
KR024707    ILARWSSFKK  NGAIKVLRGF  KKEISSMLNI  MNRRKRSVTM  LLMLLPTALA
KT187563    ILARWSSFKK  NGAIKVLRGF  KKEISSMLNI  MNRRKRSVTM  LLMLLPTALA
KP723473    ILARWSSFKK  NGAIKVLRGF  KKEISSMLNI  MNRRKRSVTM  LLMLLPTALA
JQ917404    ILARWSSFKK  NGAIKVLRGF  KKEISSMLNI  MNRRKRSVTM  LLMLLPTALA
JN713897    ILARWSSFKK  NGAIKVLRGF  KKEISSMLNI  MNRRKRSVTM  LLMLLPTALA

            ....|....|  ....|....|  ....|....|  ....|....
                110         120         130
KJ954284    FHLTTRGGEP  HMIVSKQERG  KSLLFKTSAG  INMCTLIAM
KP406801    FHLTTRGGEP  HMIVSKQERG  KSLLFKTSAG  VNMCTLIAM
EF127001    FHLTTRGGEP  HMIVSKQERG  KSLLFKTSAG  VNMCTLIAM
DQ285562    FHLTTRGGEP  HMIVSKQERG  KSLLFKTSAG  VNMCTLIAM
JQ922545    FHLTTRGGEP  HMIVSKQERG  KSLLFKTSAG  VNMCTLIAM
JN903581    FHLTTRGGEP  HMIVSKQERG  KSLLFKTSAG  VNMCTLIAM
KM403635    FHLTTRGGEP  HMIVSKQERG  KSLLFKTSAG  VNMCTLIAM
KR024707    FHLTTRGGEP  HMIVSKQERG  KSLLFKTSAG  VNMCTLIAM
KT187563    FHLTTRGGEP  HMIVSKQERG  KSLLFKTSAG  VNMCTLIAM
KP723473    FHLTTRGGEP  HMIVSKQERG  KSLLFKTSAG  VNMCTLIAM
JQ917404    FHLTTRGGEP  HMIVSKQERG  KSLLFKTSAG  VNMCTLIAM
JN713897    FHLTTRGGEP  HMIVSKQERG  KSLLFKTSAG  VNMCTLIAM
```

**Figure 4  Alignment of deduced amino acid sequence of Dengue serotype 1 (DENV 1).** The Amino acid sequence derived from DENV 1 specific amplified region (GenBank accession no. KJ954284) containing 139 amino acid residues were aligned with the other eleven DENV 1 amino acid sequences (GenBank accession no. KP406801, EF127001, DQ285562, JQ922545, JN903581, KM403635, KR024707, KT187563, KP723473, JQ917404, JN713897) of various strains retrieved from GenBank.The gene accession number of individual isolate are shown on the left. The alignment was edited with BioEdit sequence alignment editor. A mutation was observed in the test sample, where Valine was replaced with Isoleucine at 131st position and it was highlighted in the image.

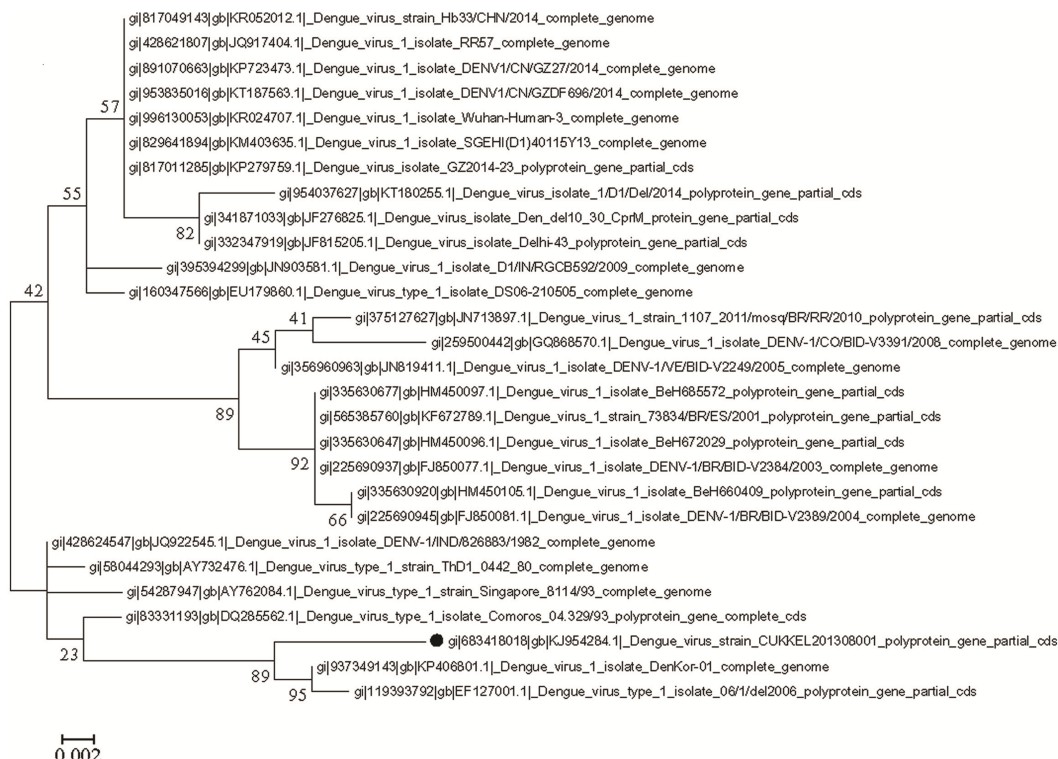

**Figure 5** **Molecular phylogenetic analysis of DENV 1 by Maximum Likelihood method, Tamura-Nei model.** The phylogenetic tree was constructed from the gene sequences of DENV isolates of different geographical locations and DENV 1 isolate (GenBank accession no. KJ954284) of northern Kerala. The bootstrap support value indicates that the sequence of dengue virus isolate from Northern Kerala shares a close relationship with DENV 1 isolate (GenBank accession no. KP406801) and DENV 1 isolate 06/1/del2006 (GenBank accession no. EF127001). The reliability of the analysis was evaluated by a bootstrap test with 1,000 replications (MEGA 6). The analysis involved total 28 nucleotide sequences.

Whereas in New Caledonia, DENV 1 and DENV 3 viruses were isolated from 6 patients with Dengue fever in 1989 (*Laille & Deubel, 1991*). The first report of dual concurrent infection with DENV 2 and DENV 3 was observed in Chinese patients returned from Sri Lanka (*Wenming et al., 2005*). The viremic serum samples (292 in total) collected during epidemics from Indonesia, Mexico, and Puerto Rico were tested, and 16 (5.5%) cases were found to contain two or more dengue viruses by reverse transcriptase–polymerase chain reaction (*Lorono-Pino et al., 1999*).

Out of 100 NS1 positive samples, 26 (26%) were found to be containing dengue viral RNA and showed a target specific band (511 bp) on an agarose gel. The low proportion of dengue viral genome in NS1 positive serum samples may be due to loss of intact viral genome or the degradation of dengue positive RNA sense strand while handling the samples. The serotyping of the dengue viral RNA samples using nested PCR with sero specific primers revealed that all samples are concurrently infected with multiple serotypes. Different combinations of Dengue virus (DENV) concurrent infections including DENV 1 and DENV 3, 34% (nine cases out of 26 cases), DENV 2 and DENV 3, 27% (seven cases out of 26 cases), DENV 1,2 and 3, 31% (eight cases out of 26 cases) were observed. Further, it
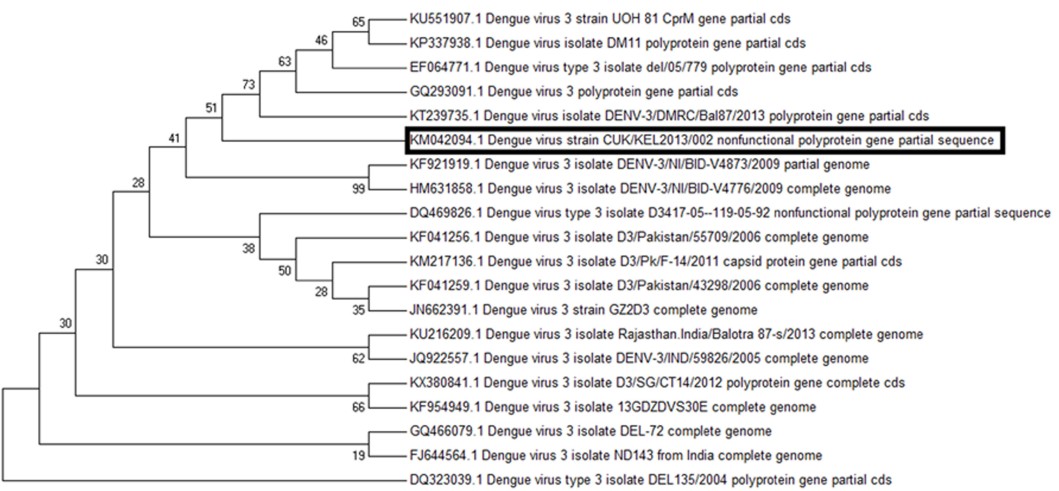

**Figure 6 Molecular phylogenetic analysis of DENV 3 by Maximum Likelihood method, Tamura-Nei model.** The phylogenetic tree was constructed based on dengue gene sequences of different geographical isolates and DENV 3 isolate obtained from northern Kerala (CUKKEL2013/02; GenBank accession no. KM042094). The analysis indicates that northern Keralian isolate is closely related to Dengue virus isolate (DENV-3/DMRC/Bal87/2013) polyprotein gene, partial cds (KT239735) of Rajasthan, India. The reliability of the analysis was evaluated by a bootstrap test with 500 replications (MEGA 7), Tamura-Nei model (*Tamura et al., 2012*).

was also observed that two of the samples were concurrently infected with all four serotypes as shown in Table 1. However, coexistence of all four serotypes in one host needs to be authenticated by characterizing the viral genome after isolation and propagation of viruses from those serum samples. The observations indicate the possibility of enhancement of future concurrent infections, as the percentage of single serotype infection decreased and concurrent infection of multiple serotypes increased based on the current investigation. As supporting evidence, there were only 19% of concurrent infections during 2006 (Delhi, India). However, there was a significant increase in the percentage of concurrent infections in 2009 (56.8%) in Ernakulum, India. The percentage of concurrent infection reached 100% based on the current investigation. It is also important to note that Kerala, a southwest state of India, provides an ideal ecosystem for the propagation of both the mosquito vectors (*Aedes albopictus* and *Aedes aegypti*) of dengue transmission.

Since the maximum nucleotide sequence was obtained in the case of DENV 1, the amino acid alignment (1-139 aa) was made with the various DENV 1 isolates existing in the data base, which revealed an identity of 98%–99%, except Isoleucine instead of Valine at 131st position as shown in Fig. 4. Since both amino acids belong to the non-polar group of standard genetic code, the amino acid change may not have any significance. Phylogenetic analysis of DENV 1 nucleotide sequences (GenBank accession no. KJ954284) obtained from 2013 epidemic samples shares a common ancestor relationship with clinical isolates of DENV 1 from South Korean Travellers (GenBank accession no. KP406801) as well as the DENV 1 isolate from Delhi, India (GenBank accession no. EF127001) in among 28 nucleotide sequences used for the analysis (Fig. 5). Similarly the phylogenetic analysis (Fig. 6) of dengue 3 virus isolate of northern Kerala indicates the close relationship to dengue

virus isolate (DENV-3/DMRC/Bal87/2013) polyprotein gene, partial cds (KT239735) of Rajasthan, India. Concurrent infection of multiple serotypes of dengue poses a lot of questions regarding viral replication. The possibility of mutual interference among different dengue serotypes during replication within the same host need to be investigated in detail. If so, would there be any replicative advantage to one of the serotypes against the other three serotypes? The viral interference which leads to replicative advantage towards one particular serotype is a major concern in the development of tetravalent dengue vaccine (*Anderson et al., 2011*).

## CONCLUSIONS

The present study shows that all samples which were able to show PCR amplified product, analysed from the northern Kerala, India between 2013 and 2015 harbour more than one serotype of dengue, indicating 100% concurrent infection. The occurrence of concurrent infection of DENV 1 and DENV 3 was higher as compared to other combinations. The DENV 1 isolates of Northern Kerala was more closely related to South Korean and Delhi strain based on phylogenetic analysis. Similarly DENV 3 was more closely related to Dengue 3 isolates of Rajasthan, India (KT239735).

## ACKNOWLEDGEMENTS

We thank Mr. Sabastian, Director at the division of Kanhangad diagnostic centre for providing the clinical samples for research purposes. We also thank all the medical staff and administrative staff of IAD (Institute of Applied Dermatology), Uliyathadka, Kasargod District, Kerala who have kindly assisted in sample processing.

### Funding

This work was supported by Department of Science and Technology (DST)- Science and Engineering Research Board (SERB), Young Scientist Scheme (SB/YS/LS-366/2013), Govt of India, New Delhi and Kerala State Council for Science, Technology & Environment (KSCSTE)-Kerala Biotechnology Commission (KBC) under Young Investigator Scheme (015/YIPBIKBC/2013/CSTE), Govt of Kerala, India. The funders had no role in study design, data collection and analysis, decision to publish, or preparation of the manuscript.

### Grant Disclosures

The following grant information was disclosed by the authors:
Department of Science and Technology (DST)-Science and Engineering Research Board (SERB): SB/YS/LS-366/2013.
Kerala State Council for Science, Technology & Environment (KSCSTE)-Kerala Biotechnology Commission (KBC): 015/YIPBIKBC/2013/CSTE.

### Competing Interests

The authors declare there are no competing interests.

## Author Contributions

- Manchala Nageswar Reddy conceived and designed the experiments, performed the experiments, analyzed the data, contributed reagents/materials/analysis tools, wrote the paper, prepared figures and/or tables, reviewed drafts of the paper.
- Ranjeet Dungdung contributed reagents/materials/analysis tools, prepared figures and/or tables.
- Lathika Valliyott reviewed drafts of the paper.
- Rajendra Pilankatta conceived and designed the experiments, wrote the paper, prepared figures and/or tables, reviewed drafts of the paper.

## Human Ethics

The following information was supplied relating to ethical approvals (i.e., approving body and any reference numbers):

Institute of Applied Dermatology (IAD), Kasaragod district, Kerala. O.R.No: IAD/IEC/13/14.

## DNA Deposition

The following information was supplied regarding the deposition of DNA sequences:

The sequences described here are accessible via GenBank accession numbers KX031992, KJ954284 and KM042094.

## Data Availability

The raw data is deposited at GenBank, and accession numbers are available in the body of the article.

## Supplemental Information

Supplemental information for this article can be found online at http://dx.doi.org/10.7717/peerj.2970#supplemental-information.

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
