# Peer review of "Occurrence of concurrent infections with multiple serotypes of dengue viruses during 2013–2015 in northern Kerala, India"

_PeerJ, doi:10.7717/peerj.2970_

## Round 0.1 · original submission · Major Revisions

Please address all the recommendations made by the reviewers.

·

Basic reporting

The study by Pilankatta and team aimed to profile the dengue virus serotypes underlying dengue infection of humans in a regional part of India. As this report appears to be one of the first studies to profile dengue serotypes in infected patients from this geographic location, it is an original study and has great potential to add scientific value to the field.

Experimental design

The authors have collected suspected dengue infected patient blood samples from hospital, confirmed infection using NS1 ELISA, and isolated viral genome for further characterization. Overall, the performed experiments are logical and required for such a study. Although the study is of great value deserving publication, its current style of presentation and some of the scientific analyses needs further improvement/explanation/experimentation.

Validity of the findings

The basic finding is that there is concurrent infection of humans by multiple dengue virus serotypes in the studied geographic location.

Additional comments

The study by Pilankatta and team aimed to profile the dengue virus serotypes underlying dengue infection of humans in a regional part of India. As this report appears to be one of the first studies to profile dengue serotypes in infected patients from this geographic location, it is an original study and has great potential to add scientific value to the field. The authors have collected suspected dengue infected patient blood samples from hospital, confirmed infection using NS1 ELISA, and isolated viral genome for further characterization. Overall, the performed experiments are logical and required for such a study.

Although the study is of great value deserving publication, its current style of presentation and some of the scientific analyses needs further improvement/explanation/experimentation. The following suggestions would make this study/manuscript more logical and scientific:

I have three main criticisms about this study:

1. The results section needs more articulation of the logic behind the each experiment, to enable the reader better understand what each experiment was meant for.

2. Why did the authors talk about only the sequencing data for DENV3 and 1 .
.why was DENV2 and 4 missing? Was it DENV1/3 given as representative?

3. A phylogeny analysis was shown only for DENV1, even though DENV3 was predominant. The authors will have to explain the reason behind showing phylogeny analysis only for DENV1 . Perhaps it would be better if they could also provide similar analysis for DENV3.

Minor comments

1. Language/grammar correction is needed
2. Isnt it that DENV is the correct abbreviation?
3. Title: 2015 in “north part” of Kerala

COMMENT: “Northern Kerala” is better?

4. “..The comparison of the amino acid sequence (1-139 aa) of DEV 1..”

COMMENT: amino acid sequence of which protein?

5.
In the last 50 years, co- circulation of dengue serotypes was reported in South Asia, including 97 developing countries such as India.

COMMENT: I am not sure how does the inclusion of “developing countries” serves the purpose, rather than just name the county alone.

6.
Both males and females belong to the 30 – 40 years age group with an approximate ratio of 2:1. This ratio is consistent with the previous reported value (Mishra et al., 2015).

COMMENT: I would replace the word “consistent” with “similar”.

Reviewer 2 ·

Basic reporting

1-The use of English language is nearly acceptable. Constructions are a bit awkward sometimes (e.g., line 83).
2-Introduction & Background:
2.1-Lines 75-81 are unrelated to the article’s subsequent content, and can be deleted.
2.2- Lines 88-89: Concurrent infections by different dengue types are quite common and, therefore, the findings are not novel. In addition to the countries mentioned by the authors, this has happened in several other regions of the world, including the same region of India as in the present report [Anoop et al, Indian J Exp Biol 48(8):849-57, 2010]. What calls some attention is the frequency of concurrent infections by
2.3- Lines 85-86: There is no firm evidence for this affirmation. There has not been solid proof that concurrent infection by multiple serotypes relates to increased disease severity. There has been abundant evidence that sequential infections by dengue virus may worsen by the presence of previous antibodies, but not that simultaneous infections have the same effect.
2.4-Lines 90-115: The authors linger for too long about co-circulation of dengue types, another observation that is present virtually worldwide. This could be reduced to a couples of sentences related specifically to what happens in India.

Experimental design

3-Materials and Methods:
3.1- This is a clinical study, so it should read ‘patients and methods’.
3.2- Lines 145-147: The text is confusing and unclear, including text on the inclusion criteria. For example, the meaning of “suffering from viremia” is unclear (one suffers from symptoms, not from viremia). Lines 148-149 state that there was no history of febrile illness during 4 weeks preceding sample collection, which is contradictory to lines 139-140.
3.3- Lines 208-213: What was the gender distribution among children?

Validity of the findings

4-Results
4.1- Lines 219-223: The frequencies of co-infections were high. All 26 samples positive for dengue virus by PCR were positive for more than one dengue type, and some of them were positive for 3 and 4 dengue virus types. Most findings were based on band sizes, with limited confirmation by sequencing. However, the method was based on nested-PCR, which is known for its propensity for cross-contamination in conventional laboratory conditions. The authors do not make clear the experimental conditions and measures taken to avoid contamination sample contamination and, most of all, cross-contamination. Considering that ALL samples positive for dengue by RT-PCR were co-infected by more than one virus type, how can they be sure that this was not artifact? Not even a single sample was positive for just one dengue virus in the whole study?
4.2- Table 1 and figure 4 show only one sequence from the present study. Where are the others? Were sequenced PCR products identical among themselves? That would strongly indicate contamination.
4.3- Lines 242-244: Another limitation of the study was that clear sequences of dengue-2 and -4 were not obtained, and therefore results for these two types relied exclusively on band sizes.
4.4- Virus isolation and plaque-purification followed by sequencing would be greatly useful to establish conclusive proof of simultaneous infection by multiple dengue virus serotypes.

5- Discussion:
5.1- Text in lines 282-290 is unnecessary.
5.2- Lines 315-324: unnecessary repetition of results.
5.3- Discussion of the low proportion of PCR-positive samples (26%) among those NS1-positive is needed, instead of just repeating text from Results.
5.4- Lines 338-340: This is quite a strong statement and a more careful tone should be used. There is not enough evidence that co-infection by more than one dengue virus type is related with severity. The authors are in an excellent place to answer that question in Kerala or elsewhere in India, since a true evaluation of severity could be carried out, with the adequate controls.
5.5- Lines 342-343: People are exposed to bites by mosquitoes individually infected with single dengue serotypes, and the existence of mosquitoes simultaneously infected by multiple dengue types, while very interesting, is not a requirement.

6- Conclusions:
6.1- Lines 362-363: Not ALL viremic samples. All samples from which authors were able to amplify product. many NS1-positive samples that were negative by this PCR method may have been viremic.
6.2- Lines 366-367: Language is unduly strong here. There is no firm evidence for that.

Additional comments

This is a provocative cross-sectional study with stron limitations in Methods. If 100% of the patients with PCR-amplifiable dengue virus in their blood (26% of 100 NS1-positive people over 3 years) are in fact simultaneously infected by several dengue types, that would indeed be very important. But the paper falls short of proving that inequivocally. Cross-contamination during nested PCR is a possibility, and sequencing os the PCR products were not done except for a few. Multiple virus types isolation and sequencing of plaque-purified viruses would be very strong. If that is not unequivocally evidenced, the paper has greatly reduced importance.

---

## Round 0.2 · accepted · Accept

The authors have properly revised the manuscript.